# Exploring the Anticancer Potential of Phenolic *nor*-Triterpenes from Celastraceae Species

**DOI:** 10.3390/ijms25179470

**Published:** 2024-08-30

**Authors:** Carolina P. Reyes, Alejandro Ardiles, Laura Anaissi-Afonso, Aday González-Bakker, José M. Padrón, Ignacio A. Jiménez, Félix Machín, Isabel L. Bazzocchi

**Affiliations:** 1Instituto Universitario de Bio-Orgánica Antonio González, Departamento de Bioquímica Microbiología, Biología Celular y Genética, Universidad de La Laguna, Av. Astrofísico Francisco Sánchez 2, 38206 La Laguna, Spain; cpreyes@ull.es; 2Departamento de Ciencias Básicas, Facultad de Ciencias, Universidad Santo Tomás, Avenida Iquique, Antofagasta 3991, Chile; aardiles2@santotomas.cl; 3Unidad de Investigación, Hospital Universitario Ntra Sra de Candelaria, Ctra del Rosario 145, 38010 Santa Cruz de Tenerife, Spain; anaissi.ull@gmail.com; 4Instituto Universitario de Bio-Orgánica Antonio González, Universidad de La Laguna, Av. Astrofísico Francisco Sánchez 2, 38206 La Laguna, Spain; agonzaba@ull.es (A.G.-B.); jmpadron@ull.es (J.M.P.); 5Instituto Universitario de Bio-Orgánica Antonio González, Departamento de Química Orgánica, Universidad de La Laguna, Av. Astrofísico Francisco Sánchez 2, 38206 La Laguna, Spain; ignadiaz@ull.es; 6Instituto de Tecnologías Biomédicas, Universidad de La Laguna, 38200 La Laguna, Spain; 7Facultad de Ciencias de la Salud, Universidad Fernando Pessoa Canarias, 35450 Las Palmas de Gran Canaria, Spain

**Keywords:** *Maytenus* sp., *Celastrus vulcanicola*, celastroloids, anticancer profile, human tumour cell lines, live cell imaging, *Saccharomyces cerevisiae* model

## Abstract

To explore new compounds with antitumour activity, fifteen phenolic *nor*-tripterpenes isolated from Celastraceae species, *Maytenus jelskii*, *Maytenus cuzcoina,* and *Celastrus vulcanicola*, have been studied. Their chemical structures were elucidated through spectroscopic and spectrometric techniques, resulting in the identification of three novel chemical compounds. Evaluation on human tumour cell lines (A549 and SW1573, non-small cell lung; HBL-100 and T-47D, breast; HeLa, cervix, and WiDr, colon) revealed that three compounds, named 6-oxo-pristimerol, demethyl-zeylasteral, and zeylasteral, exhibited significant activity (GI_50_ ranging from 0.45 to 8.6 µM) on at least five of the cell lines tested. Continuous live cell imaging identified apoptosis as the mode of action of selective cell killing in HeLa cells. Furthermore, their effect on a drug-sensitive *Saccharomyces cerevisiae* strain has been investigated to deepen on their mechanism of action. In dose-response growth curves, zeylasteral and 7α-hydroxy-blepharodol were markedly active. Additionally, halo assays were conducted to assess the involvement of oxidative stress and/or mitochondrial function in the anticancer profile, ruling out these modes of action for the active compounds. Finally, we also delve into the structure-activity relationship, providing insights into how the molecular structure of these compounds influences their biological activity. This comprehensive analysis enhances our understanding of the therapeutic potential of this triterpene type and underscores its relevance for further research in this field.

## 1. Introduction

Cancer is currently considered to be one of the most complex diseases, with an aetiology not completely elucidated [1]. There have been outstanding advances in cancer treatment in the last few decades [2], both for therapeutic and diagnosis purposes. Despite that, most of the current treatments are still based on the use of chemotherapeutic agents, which are devoid of selectivity, leading to severe side effects [3], and frequently undergo chemoresistance [4]. Therefore, there is an ongoing need for novel and safe anticancer agents. In this regard, natural products, which include curcumin, epigallocatechin 3-gallate, resveratrol, sulforaphane, and withaferin A, represent a rich source of drug leads [5], and they have contributed greatly to anticancer drug discovery and development since they are characterised by long-term safety and negligible and/or inexistent side effects having been proposed as possible adjuvants to traditional chemotherapy [6].

The yeast *Saccharomyces cerevisiae* is a model unicellular eukaryote that has proven very useful for understanding the mode of action (MoA) of toxic drugs and xenobiotics [7,8]. Indeed, yeast has served to confirm the MoA of many DNA-damaging agents widely used in cancer chemotherapy. The strategy to confirm this MoA is through chemical-genetic interactions whereby a DNA-damaging agent renders yeast DNA repair mutants (e.g., *rad* mutants) extremely hypersensitive compared to a reference wild-type strain [9]. Likewise, yeast mutants for the oxidative stress response (e.g., Δ*yap1*) are hypersensitive to oxidative stressors, a common MoA of many xenobiotics [10], which results in undesirable non-specific side effects. In addition to chemical genetics, oxidative stress can be further assessed in wild-type strains because this yeast is a facultative anaerobe, i.e., it can grow without O_2_, provided that fermentable carbon sources such as sugars are included in the growth medium [11]. On the other hand, the addition of a non-fermentable carbon source such as glycerol in normoxic conditions forces yeast to rely solely on mitochondrial respiration. Thus, a comparison of growth in fermentable versus non-fermentable media can determine whether a compound interferes with the respiratory chain.

The Celastraceae family is distributed mainly in tropical and subtropical regions of the world, including North Africa, South America, and East Asia, and their species have a long history in traditional medicine [12]. The therapeutic potential of *Maytenus* species [13] has been mainly attributed to *nor*-triterpenes with a 24-*nor*-*D:A*-friedo-oleanane skeleton, named celastroloids, which are chemotaxonomic markers of the family [14]. This class of natural products shows a wide range of bioactivities and has garnered significant interest as versatile compounds capable of modulating diverse molecular and cellular pathways [15]. Regarding the phenolic *nor*-triterpenes (PTs), some studies have been reported to exhibit cytotoxicity against human tumour cell lines [16]. Previous works report on the isolation of bioactive pentacyclic [17] and tetracyclic [18] triterpenoids from *Maytenus jelskii*, *Maytenus cuzcoina,* and *Celastrus vulcanicola*. However, no phytochemical studies on the PT content in the root bark of these species have been performed.

In order to explore the antitumoral potential of metabolites from Celastraceae species, fifteen phenolic *nor*-triterpenes isolated from *M. jelskii*, *M. cuzcoina,* and *C. vulcanicola* have been studied. Their chemical structures were elucidated through spectroscopic and spectrometric techniques, resulting in the identification of three novel chemical entities and three ones reported for the first time as natural products. Their biological screening against a panel of six human solid tumour cell lines (A549, HBL-100, HeLa, SH-SY5Y, T-47D, and WiDr) revealed that three compounds exhibited remarkable activity and were selected to deepen into their mode of action through continuous live cell imaging. Furthermore, their effects on a drug-sensitive *Saccharomyces cerevisiae* strain were investigated. Herein, it also delves into the structure-activity relationship, providing further insights into the therapeutic potential of this type of natural compound.

## 2. Results

### 2.1. Chemistry

In this study, three Celastraceae species, *Maytenus jelskii*, *Maytenus cuzcoina,* and *Celatrus vulcanicola,* were studied for their content of phenolic *nor*-triterpenes (PTs). Multiple chromatographic steps of root bark extracts of the plants under study yielded three previously undescribed PTs (**1**, **4,** and **6**), along with three ones reported for the first time as natural products (**2**, **3,** and **5**) and nine known PTs (**7**–**15**) (Figure 1). Their structures were greatly aided by NMR techniques (Appendix A), including two-dimensional NMR spectra (COSY, ROESY, HSQC, and HMBC), and a comparison of their spectroscopic data with those previously reported.

The known PTs were identified as 6β-acetonyl-tingenol (**2**) [19], 6β-acetonyl-22β-hydroxy-tingenol (**3**) [19], 6β-acetonyl-20β–hydroxy-tingenol (**5**) [19], 3-*O*-methyl-23-hydroxy-6-oxo-tingenol (**7**) [20], cognatine (**8**) [21], 6-oxo-pristimerol (**9**) [22], 7-hydroxy-6-oxo-pristimerol (**10**) [23], 23-*nor*-6-oxo-pristimerol (**11**) [24], demethyl-zeylasteral (**12**) [25], blepharotriol (**13**) [25], zeylasteral (**14**) [26], and 7α-hydroxy-blepharodol (**15**) [26] by NMR data and comparison with those reported in the literature. The structures of the novel phenolic *nor*-triterpenes (**1**, **4,** and **6**) were elucidated as described below.

Compound **1** was isolated as an amorphous solid with a molecular formula of C_33_H_46_O_5_ calculated by HREIMS and showed bands in its IR spectrum for hydroxyl (3418 cm^−1^) and carboxyl groups (1714 cm^−1^). The ^1^H NMR spectrum (Table 1, Appendix A) showed signals assigned to seven methyl groups as singlets, five angular ones (δ_H_ 0.52, 1.05, 2 × 1.17, and 1.45), a methyl in α position to a carbonyl group (δ_H_ 2.16) and one methyl on the aromatic ring (δ_H_ 2.18). In addition, two singlet signals were assigned to a methoxy group at δ_H_ 3.52 and an aromatic proton at δ_H_ 6.81. Signals corresponding to two methine protons at δ_H_ 5.85 (d, *J* = 6.2 Hz) and δ_H_ 3.99 (ddd, *J* = 2.5, 6.2, 10.4 Hz), and two methylene protons at δ_H_ 2.71 (dd, *J* = 2.5, 16.7 Hz) and δ_H_ 2.45 (d, *J* = 10.4, 16.7 Hz) in an AMXY spin system were also observed in the ^1^H NMR spectrum and assigned to H7, H6 y H_2_1′ protons, respectively. Moreover, signals at δ_H_ 5.55 and 6.63, interchangeable with D_2_O suggest two hydroxyl groups in the molecule.

The ^13^C NMR spectrum (Table 1, Appendix A) showed signals for 33 carbons, including signals characteristic of a methyl ester group (δ_C_ 51.5 and 179.3), eight carbon resonances between δ_C_ 109.2 and 150.1 confirmed the aromatic ring and double bond, and a ketone group a δ_C_ 208.1. These data and 2D NMR experiments (COSY, ROESY, HSQC, and HMBC) indicated compound **1** was a phenolic *nor*-triterpene related to the previously reported 6-oxo-pristimerol (**9**) [22], also isolated in the present study. The most noteworthy differences in their ^1^H NMR spectra were the presence of the AMXY system (H7, H6, and H_2_1′), the methyl group at δ_H_ 2.16, and the shift of the Me23 from δ_H_ 2.78 in the 6-oxopristimerol to δ_H_ 2.18 in **1**, indicating Me23 is not in a *sin*-periplanar position to a carbonyl on C-6. These data suggest an acetonyl group in the molecule, which was confirmed by a COSY experiment, showing correlations of H6/H7 protons with the methylene protons H_2_1′.

The regiosubstitution pattern in the molecule was achieved by an HMBC experiment (Figure 2a). Thus, the acetonyl group was located on C6 by correlation of H6 (δ_H_ 3.99) with C1′ (δ_C_ 51.9), C7 (δ_C_ 121.7), C5 (δ_C_ 128.1), C10 (δ_C_ 142.0), and C8 (δ_C_ 150.1). In addition, long-range correlations were observed from H1 (δ_H_ 6.81) to C3 (δ_C_ 140.3) and C9 (δ_C_ 37.2), and those between Me30 (δ_H_ 1.17) and C29 (δ_C_ 179.3). The α stereochemistry of H6 was determined by the constant couplings and a Roesy experiment (Figure 2b), showing a cross-peak between H1′ and Me25, indicating a β-disposition of the acetonyl group. Moreover, an α-disposition of the methoxy carbonyl ester was assigned by the observed correlation between Me27 (δ_H_ 0.52) and the Me ester at δ_H_ 3.52. All these data established the structure of compound **1** as 6β-acetonyl-pristimerol.

The structure of compound **4** was established by means of NMR techniques, including 2D experiments (COSY, ROESY, HSQC, and HMBC), IR, UV, and mass spectroscopy. Thus, **4** showed the molecular formula of C_31_H_42_O_5_, calculated by HREIMS, and bands in its IR spectrum for hydroxyl (3620–3218 cm^−1^) and carbonyl groups (1708 cm^−1^). The ^1^H and ^13^C NMR spectra (Table 1 and Appendix A) indicated that **4** was a phenolic *nor*-triterpene with a tingenone [27] skeleton related to **1**. The most significant differences in their structures are the substituents on ring E. Thus, signals at δ_H_ 2.16 d (*J* = 19.4 Hz) and 2.69 d (*J* = 19.4 Hz) in the ^1^H NMR spectrum, characteristic of protons in α to a carbonyl group (H22), together with signals corresponding to a quaternary carbon linked to a hydroxyl group (C20, δ_C_ 74.2) and a carbonyl carbon signal at δ_C_ 215.5 in its ^13^C NMR spectrum, established the structure of compound **4**. An HMBC experiment showing three-bond correlations of the signal assigned to Me30 (δ_H_ 1.40) with the carbon resonances δ_C_ 34.2 (C19), 72.3 (C20), and 217.0 (C21) allowed to establish the regiosubstitution of the tertiary alcohol and the aliphatic ketone in the molecule. Moreover, the α stereochemistry of the tertiary hydroxyl group on C20 was determined by a Roesy experiment, showing cross-peak of Me27 (δ_H_ 0.59) with H22α (δ_H_ 2.69) and a correlation from Me30 (δ_H_ 1.40) to H18 (δ_H_ 1.69) and Me28 (δ_H_ 1.28). These data are in accordance with the structure of compound **4** as 6β-acetonyl-20α-hydroxy-tingenol.

The spectroscopic and spectrometric data for compounds **2**, **3,** and **5** were coincident with those previously reported for 6β-acetonyl-tingenol, 6β-acetonyl-22β-hydroxy-tingenol, and 6β-acetonyl-20β-hydroxy-tingenol, respectively. However, these reported compounds were acid-catalysed products in acetone from the natural *nor*-triterpene methylene quinones, tingenone III, 22β-hydroxy-tingenone and 20-hydroxy-20-*epi*-tingenone [19]. In the present work, compounds **2**, **3,** and **5** are reported for the first time as natural products. Since during the extraction process of these compounds, neither acetone nor concentrated HCl was used, we assumed they were natural products, and not artefacts formed during the extraction.

The biosynthesis of 1 could be explained by a conjugated addition of the ketoacetyl-coenzyme A to C6 of the conjugated quinone system, leading to the aromatisation of the A-ring [28] (Figure 3). Furthermore, the biosynthesis of compounds **2**–**5** could be proposed by 1,6-conjugated addition of ketoacetyl-coenzyme A to tingenone [27] to give **2**, whereas successive enzymatic transformations lead to metabolites **3**, **4,** and **5**.

Compound **6** exhibited a molecular formula of C_31_H_40_O_6_ calculated by HREIMS and ^13^C NMR spectrum. The IR spectrum exhibited bands corresponding to hydroxyl (3391 cm^−1^) and ester (1724 cm^−1^) groups. Its ^1^H NMR spectrum (Table 1 and Appendix A) showed signals corresponding to six methyl groups, including a methyl on double bond at δ_H_ 1.72 and one methyl on the aromatic ring at δ_H_ 2.73. In addition, signals assigned to two methoxy groups (δ_H_ 3.72 and 3.79), an aromatic proton (δ_H_ 6.92), a proton in α position to an α,β-unsaturated ketone (δ_H_ 6.00, s), and a geminal proton to secondary alcohol at δ_H_ 4.20 (dd, *J* = 5.4, 11.6 Hz) were observed in the ^1^H NMR spectrum. A single signal at δ_H_ 6.18, interchangeable with deuterated water, indicated a hydroxyl proton. The ^13^C RMN spectrum (Table 1 and Appendix A) revealed 31 carbon resonances, including those characteristic for an α,β-unsaturated carbonyl at δ_C_ 186.1, and ten olefinic carbons, including six olefinic carbons (δ_C_ 110.2 d; 122.0 s; 134.2 s; 144.7 s; 151.6 s and 151.9 s) and four vinylic ones (δ_C_ 127.5 d; 128.8 s; 134.9 s and 161.5 s). A methine carbon linked to an oxygen atom at δ_C_ 68.1 and two methoxy resonance carbons at δ_C_ 52.2 (q) and 61.1 (q) were also observed. All these data are in accordance with the structure of a phenolic *nor*-triterpene related to the previously reported one, cognatine [21], also isolated (compound **8**) in the present study from *Maytenus jelskii*.

The regiosubstitution pattern in the molecule was determined by an HMBC experiment. Thus, the correlation between the signal assigned to H1 and the aromatic quaternary carbons C3, C10, and C5, the correlation from the methoxy group at δ_H_ 3.72 with C3, and those from the Me23 with C3, C4, and C5, established the sub-structure of A ring in the molecule. Moreover, the observed correlations of Me25 (δ_H_ 1.37) with C10 and C8, and those of H7 (δ_H_ 6.00) with carbon resonances assigned to C5 and C9 confirmed the α,β-unsaturated ketone on ring B. The double bond was located on ∆^14^ by the long-range correlations from Me27 at δ_H_ 0.87 to C13 and C14 and those from Me26 at δ_H_ 1.72 to C14 and C16, whereas correlations of H19α (δ_H_ 1.89), H21 (δ_H_ 4.20), Me30 (δ_H_ 1.21) and OMe (δ_H_ 3.79) with C29 (δc 178.9) established the regiosubstitution on the E ring. The β-equatorial stereochemistry of the secondary hydroxyl group and α disposition of the C29-methoxy group were deduced from a ROESY experiment, showing cross-peaks between H21/Me30 and OMe29/Me27, respectively. All these data established the structure of compound **6** as 22-deoxo-cognatine.

### 2.2. Biological Assays

#### 2.2.1. Antiproliferative Activity on Cancer Cell Lines

The in vitro antiproliferative activity of the isolated compounds (**1**–**15**) was evaluated against six human tumour cell lines: A549 (non-small cell lung), HBL-100 (breast), HeLa (cervix), SW1573 (non-small cell lung), T-47D (breast), and WiDr (colon) [29]. The results (Table 2) revealed that compounds **9** (6-oxo-pristimerol), **12** (dimethyl-zeylasteral), and **14** (zeylasteral) showed remarkable activity on all tested cell lines, exhibiting GI_50_ values <10 μM. Compound **9** exhibited GI_50_ values of 3.1 and 4.0 μM on HBL-100 and SW1573 cell lines, respectively, whereas compound 14 showed significative activity on HBL-100, HeLa, and SW1573 cell lines with GI_50_ values of 3.4, 4.1, and 3.0 μM, respectively.

Noteworthy, compound **12**, the most active compound, showed more potency than the two reference drugs, cisplatin, and 5-fluouracil, exhibiting GI_50_ values ranging from 0.4 to 2.4 μM on the assayed cell lines, except for WiDr cell line (GI_50_ 5.8 μM). Moreover, compounds **1**–**4**, **8**, **10**, **11**, **13,** and **15** showed weak activity (GI_50_ 10–25 μM), whereas compounds **5**, **6,** and **7** were inactive (GI_50_ > 25 μM) on all the assayed cell lines.

#### 2.2.2. Structure-Activity Relationship Analysis

The influence of substitution patterns of the phenolic *nor*-triterpenoids on their antiproliferative activity (Table 2) was analysed considering the triterpene scaffold, netzahualcoyone [14], tingenone [27] or pristimerin [30], and substituents on A, B, or E rings. This analysis revealed the following trends on the structure-activity relationship (SAR): (a) Compounds with a tingenone (**2**, **3**, **4**, **5**, and **7**) or netzahualcoyone (**6** and **8**) skeleton were inactive, whereas those with a pristimerin scaffold (**1**, **9**–**15**) present a wide range of activities from inactive to potent. (b) The best substituent on the A ring for activity was the formyl group at C4, as deduced by a comparison of the activity of **7** (CH_2_OH), **9** (CH_3_), **13** (OH), and **14** (CHO). (c) Regarding the substituents on the B ring, a 6-alkyl side chain at C6 (compound **1**) or a hydroxyl group on C7 (**10**) have a detrimental effect on activity, being the best moiety of the ketone α,β-unsaturated (compounds **9**, **12,** and **14**). (d) The stereochemistry plays a crucial role in the activity, as the inversion of configuration at C-20 in the E ring (compound **4**) led to a loss of activity in **5**. Moreover, the biological profile from compounds **12** and **14** indicated that the carboxylic acid group at C29 is the best substituent, which would be explained by possible hydrogen bond interactions with the receptor and/or hydrophilicity.

In summary, the most promising anticancer compounds, **12** and **14**, are PTs bearing an aldehyde group at C4, indicating this moiety is a favourable trend for optimal activity against the tested cell lines. Moreover, the activity is increased for the compound having a carboxyl group instead of a methyl ester at C29, as could be deduced comparing **12** vs. **14**, being compound **12** more potent against all the assayed tumour cell lines than the standard anticancer drugs, cisplatin (CDDP) and 5-fluorouracil (5-FU), used as positive controls. However, further tests with non-tumour cell lines are required to uncover whether these compounds are also as selective as CDDP and 5-FU.

#### 2.2.3. Getting Insights on the Mode of Cell Death

The analysis through continuous live cell imaging provided insights into the antiproliferative effects of the compounds in the cells (Appendix A). Cell death is a process that rarely occurs as an instant event. With this technique, it is possible to observe the predominant mode of death as well as to analyse the potency of the compounds, in addition to cell death kinetics. For this study, HeLa cells were exposed to compounds **9**, **12,** and **14** at a dose of 25 μM for 20 h. The recording took place at intervals of 10 min. Figure 4 shows representative snapshots taken at different time points. The three compounds under study induced cell death by apoptosis, but at different times of exposure. Evident apoptosis hallmarks observed in treated cells included cell shrinkage, nuclear condensation, and subsequent fragmentation (Figure 4). For compound **14**, apoptosis was notorious after 5 h of exposure, while for compound **12,** the first apoptotic cells appeared later (8–10 h after exposure). In contrast, apoptosis induced by compound **9** started 10 h after exposure. In addition, after 20 h of exposure, cells treated with **12** and **14** were all dead. This was not the case with cells treated with compound **9**, where only half of the population appeared dead.

Continuous live cell imaging experiments gave additional information, indicating that the compounds affected cells at diverse speed rates. The segmentation analysis allowed supporting these observations. Figure 5 shows the kinetics of dry mass density (DMD) and total dry mass (TDM). One of the apoptosis hallmarks is the collapse of cells, which produces cell shrinkage and consequently, DMD increases. Changes over time on DMD (Figure 5A) show that cells exposed to compound **12** started to die at around 8 h after exposure, after 12 h for cells treated with compound **14**. In contrast, cells exposed to compound **9** did not show variability in DMD when compared to untreated cells. TDM is a parameter that is proportional to the number of existing cells. Thus, an increase in TDM is a sign of cell proliferation. The kinetics of TMD show that compound **9** did not prevent the growth of the cell population, as denoted by a constant increase in TMD. However, this relevant increase in TMD for cells exposed to compounds **12** and **14** was not observed.

#### 2.2.4. Determination of Toxicity, DNA Damage, Oxidative Stress and Drug Resistance Profiles in Yeast

After confirming that some of the PTs evaluated showed potent activity against cancer cell lines, the effect of the isolated PTs on *Saccharomyces cerevisiae* strains was investigated to deepen their mode of action. Chemical-genetic interactions were determined by comparing the toxicity profiles in the reference wild-type strain BY4741 and two isogenic mutants for the oxidative (Δ*yap1*) and DNA damage response (Δ*rad9* Δ*rad52*; ΔΔrad), respectively. Dose-response growth inhibition assays showed that these strains were mostly insensitive to all compounds (Figure 6). Since a sigmoidal decline in growth within the concentration range tested (1 to 128 µM) was not observed, a proper GI_50_ could not be obtained. Thus, we decided to determine the relative aggregate growth (RAG; equivalent to the area under the curve) to uncover subtle differences among the compounds. This parameter is normalised to the vehicle (DMSO) and gives a value between 0 and 1, which is further weighed to the concentrations where inhibition occurs [31]. Thus, a value of 0 implies total inhibition in the range (GI_50_ < 1 µM), whereas a value of 1 implies no inhibition whatsoever in the whole range (GI_50_ > 128 µM). RAG can also measure nonsigmoidal partial inhibition. However, 14 out of the 15 compounds yielded RAGs of 1 for the three strains (Figure 6A,B). The sole exception was compound **12**, the most active compound against tumour cell lines (Table 2), which yielded an RAG of ~0.75 due to a partial growth inhibition observed at the maximum concentration (Figure 6C). Since the inhibition was the same in the three strains, it was concluded that **12** exerts its toxicity through a MoA that is independent of DNA damage and oxidative stress. Lastly, halo inhibition assays confirmed the low toxicity of the compounds in the three strains (Figure 6D).

The high Insensitivity of *S. cerevisiae* to this set of compounds led us to test them against a mutant strain for the most important pleiotropic drug resistance (ΔΔΔΔpdr) mechanisms [32]. This drug-sensitive strain lacks three critical transcription factors (Pdr1, Pdr3, and Yrr1) that control the expression of multiple genes involved in drug resistance, as well as a constitutive efflux pump (Yor1). In contrast to the BY4741 strain, the ΔΔΔΔpdr strain was sensitive to 11 out of 15 compounds in the 1–128 µM dose-response curve (Figure 7A); only 4, **7**, **9,** and **10** remained non-cytotoxic. A group of seven compounds exhibited similarly good activities (RAG ~0.2; GI_50_ ~10–20 µM), and in it, **12** were included (**1**, **2**, **3**, **6**, **11**, **12,** and **13**) (Figure 7B,C). Two additional compounds (**5** and **8**) showed slightly lower activities, whereas two others turned out to be the most potent, even stronger than **12** (**14** and **15**). Assuming a model of two components in this cell-based assay, one component being the intracellular target and the other one representing the degree to which the drug overcomes the drug resistance mechanisms, we can conclude that **14** and **15** are likely the best hits, but **12** represents the optimal balance between target potency and accessibility.

In addition to the comparative dose-response cytotoxic profile, the ΔΔΔΔpdr strain was used to test MoAs related to oxidative stress and/or respiratory chain integrity. To this end, halo assays were performed under conditions that force growth by either fermentation (anoxia) or respiration (a non-fermentable carbon source; glycerol in this case) (Figure 7D). In the reference growth conditions, in which growth occurs through a combination of fermentation and respiration, the halo profile generally fits well with the dose-response curves, with **14** and **15** being the most toxic compounds. Interestingly, compound **6** showed a halo profile under “respiration only” compatible with disturbances of the respiratory chain. Other compounds showed neither diminished growth inhibition in anoxia nor more inhibition in glycerol, thus ruling out the contribution of these MoAs to their cytotoxicity profile.

#### 2.2.5. Determination of Antibacterial Activity

Apart from yeast, 1 to 128 µM dose-response growth curves were performed against three pathogenic bacteria: the Gram-negative *Escherichia coli* ATCC35218 strain, the Gram-positive *Enterococcus faecalis* ATCC29212 strain, and the *Staphylococcus aureus* multiresistant NRS402 strain [33]. There was no growth inhibition in the concentration range (Appendix A). This result points out that active compounds must have an eukaryotic-specific target. In addition, and together with the MoA data obtained with *S. cerevisiae* above, this suggests that these compounds are not promiscuous, thereby highlighting their potential for development as drugs.

## 3. Materials and Methods

### 3.1. General Experimental Procedures

Optical rotations were measured on a Perkin Elmer 241 automatic polarimeter in CHCl_3_ at 20 °C, and the [α]_D_ values are given in 10^−1^ deg cm^2^/g. UV spectra were obtained in absolute EtOH on a JASCO V-560 spectrophotometer. IR (film) spectra were measured on Bruker IFS 55 spectrophotometer. ^1^H (500 MHz) and ^13^C NMR (125 MHz) spectra were recorded at 300 K on the Bruker Avance 500 spectrometer; the chemical shifts are given in δ (ppm) with residual CDCl_3_ (δ_H_ 7.26, δ_C_ 77.0) as internal reference and coupling constants in Hz; COSY, ROESY (spin lock field 2500 Hz), HSQC, and HMBC (optimised for *J* ¼ 7.7 Hz) experiments were carried out with the pulse sequences given by Bruker. EIMS and HREIMS were obtained on a Micromass Autospec spectrometer. Silica gel 60 (particle size of 15–40 and 63–200 mm) and Sephadex LH-20 (Pharmacia Biotech) were used for column chromatography (CC), while silica gel 60 F254 was used for analytical or preparative thin layer chromatography (TLC), and nanosilica gel 60 F254 for high-performance TLC (HPTLC). Centrifugal planar chromatography (CPC) was performed by a Chromatotron instrument (model 7924T, Harrison Research Inc., Palo Alto, CA, USA) on manually coated silica gel 60 GF254 (Merck) using 1, 2, or 4 mm plates. The developed TLC plates were visualised by UV light and then by spraying with a staining system of H_2_O–H_2_SO_4_–HOAc (1:4:20), followed by heating of silica gel plates to approximately 150 °C.

### 3.2. Plant Material

*Maytenus jelskii* Zahlbr. (Celastraceae) was collected in Pumahuanca (latitude: 13°070′ S, longitude: 072°030′0000″ W, elevation: 3200 m), Cuzco, Perú in December 2006. The plant was identified by Alfredo Tupayachi Herrera from the Universidad Nacional de San Antonio Abad del Cusco (Perú), and a voucher specimen (CUZ 29845) was deposited at the Herbarium of the Missouri Botanical Garden, St. Louis, MO, USA.

*Maytenus cuzcoina* Loes. (Celastraceae) was collected at Huayllabamba-Urquillos (latitude: 13°210′1500″ S, longitude: 072°030′5500″ W, elevation: 2970 m), province of Urubamba, Cusco (Peru), in December 1998, and was identified by Professor. Alfredo Tupayachi Herrera from the Universidad Nacional de San Antonio Abad del Cusco (Perú). A voucher specimen (cuz 02765 A.T. 1004 MO) was deposited in the herbarium of Vargas, Department of Botany, in the Universidad Nacional de San Antonio Abad del Cusco, Perú.

*Celastrus vulcanicola* Donn. Sm. (Celastraceae) was collected in June 2004 at the Montecristo National Park (latitude: 14°210′0000″ N, longitude: 089°240′0000″ W, elevation: 2400 m), Province of Santa Ana, El Salvador. The plant was identified by Jorge Alberto Monterrosa Salomon, Curator of the Herbarium at the Jardín Botanico, La Laguna, El Salvador. A voucher specimen (J. Monterrosa and R. Carballo 412) is deposited in the Herbarium of the Missouri Botanical Garden, St. Louis, MO, USA.

### 3.3. Extraction and Isolation

#### 3.3.1. *Maytenus jelskii*

The root bark (750 g) of *M. jelskii* was extracted with hexanes–Et_2_O (4 L) in a Soxhlet apparatus. Evaporation of the solvent under reduced pressure yielded 60.3 g residue, which was chromatographed on a silica gel column using increasing polarity mixtures of hexanes–EtOAc as eluant to afford 72 fractions. Fractions 41–48 (4.1 g) were subjected to column chromatography over Sephadex LH-20 (hexanes–CHCl_3_–MeOH, 2:1:1) and silica gel (hexanes–EtOAc, and CH_2_Cl_2_–acetone of increasing polarity). Preparative HPTLC developed with hexanes-Et_2_O (6:4) was used to afford the new compound **1** (18.0 mg). Fractions 49–65 (5.0 g) were subjected to silica gel CC and eluted with a gradient of hexanes–EtOAc mixtures, and preparative HPTLC with mixtures of hexanes–Et_2_O (2:8) gave rise to the new compound **6** (4.5 mg), in addition to the known phenolic *nor*-triterpenes (PTs), **7** (7.5 mg, 3-*O*-methyl-23-hydroxy-6-oxo-tingenol), **8** (2.2 mg, cognatine) [21], **9** (3.1 mg, 6-oxo-pristimerol) [22], **10** (2.3 mg, 7-hydroxy-6-oxo-pristimerol) [23], **11** (1.1 mg, 23-*nor*-6-oxo-pristimerol) [24], **13** (2.5 mg, blepharotriol) [25] and **14** (4.1 mg, zeylasteral) [26]. Fractions 66–72 (2.9 g) were subjected to silica gel CC and eluted with a gradient of hexanes–EtOAc mixtures, and preparative HPTLC with mixtures of hexanes–Et_2_O (2:8) yielded the known PTs, **12** (demethyl-zeylasteral) [25] and **15** (7α-hydroxy-blepharodol) [26].

#### 3.3.2. *Maytenus cuzcoina*

The root bark (0.9 kg) was extracted with hexanes–Et_2_O in a Soxhlet apparatus, and after evaporation of the solvent, the extract (25.5 g) was chromatographed on Sephadex LH-20 (hexanes–CHCl_3_–MeOH, 2:1:1) to afford 64 fractions, which were combined in nine fractions (A–I) on the basis of their TLC profiles. Preliminary NMR studies revealed that fraction G showed characteristic pattern signals of PTs and were further investigated. Thus, fraction G was purified by silica gel flash column chromatography and eluted in a step gradient manner with CH_2_Cl_2_–Me_2_CO (10:1 to 8:2) to provide combined G1-G6 sub-fractions on the basis of their TLC profiles. Sub-fraction G4 was further purified by preparative TLC (CH_2_Cl_2_–Me_2_CO, 8.5:1.5) to yield the new compound **6** (7.5 mg) and **1** (13.0 mg), together with the known ones cognatine (**8**, 12.5 mg) [21], 6-oxo-pristimerol (**9**, 22.5 mg) [22], 7-hydroxy-6-oxo-pristimerol (**10**, 16.1 mg) [23], and 7α-hydroxy-blepharodol (**15**, 14.7 mg) [26].

#### 3.3.3. *Celastrus vulcanicola*

The air-dried and powdered root bark of *C. vulcanicola* (0.8 kg) was extracted with hexanes–Et_2_O (1:1, 4.0 L, 48 h) in a Soxhlet apparatus. Briefly, evaporation of the solvent under reduced pressure provided the crude extract (24.0 g), which was subjected to column chromatography over silica gel (1.0 kg) by using increasing polarity mixtures of hexanes-EtOAc (0–100%) as eluent to afford 34 fractions, which were combined into nine fractions (V1–V9). Fraction V7 (1.1 g) was separated by chromatography on Sephadex LH-20 (hexanes–CHCl_3_-MeOH, 2:1:1), on silica gel (hexanes-EtOAc of increasing polarity, 8:2 to 3:7), by CPC centrifugal planar chromatography (CH_2_Cl_2_–Me_2_CO of increasing polarity, 9:1 to 7:3) and by preparative HPTLC (hexanes–Et_2_O, 3:7) to give the new compound **4** (10.5 mg), in addition to the known PTs, 6β-acetonyl-tingenol (**2**, 14.5 mg) [19], 6β-acetonyl-22β-hydroxy-tingenol (**3**, 8.0 mg) [19], 6β-acetonyl-20β–hydroxy-tingenol (**5**, 3.1 mg) [19], 6-oxo-pristimerol (**9**, 3.2 mg) [22], 23-*nor*-6-oxo-pristimerol (**11**, 2.8 mg) [24], demethyl-zeylasteral (**12**, 1.4 mg) [25] and zeylasteral (**14**, 1.5 mg) [26].

#### 3.3.4. Spectroscopic Data

6β-Acetonyl-pristimerol (**1**). Amorphous solid; [α]_D_^20^ −36.4 (*c* 1,02, CHCl_3_); UV (EtOH) λ_max_ (log є) 314 (3,1), 284 (3,4), 262 (3,4) nm; IR ν_max_ 3418, 2946, 2870, 1714, 1618, 1506, 1486, 1455, 1290, 1216, 1157, 1019, 756 cm^−1^; ^1^H NMR (CDCl_3_, 500 MHz) δ 2.16 (3H, s, Me3′), 2.45 (1H, dd, *J* = 10.4, 16.7 Hz, H1′), 2.71 (1H, dd, *J* = 2.5, 16.7 Hz, H1′), 3.52 (3H, s, OMe), 5.55 (1H, s, OH), 6.63 (1H, s, OH), for other signals, see Table 1; ^13^C NMR (CDCl_3_, 125 MHz) δ 30.5 (q, Me3′), 51.5 (q, OMe), 51.9 (t, C1′), 208.1 (s, C2′), for other signals, see Table 1; EIMS *m/z* (%) 522 (M^+^, 6), 464 (16), 253 (15), 243 (20), 241 (36), 215 (22), 203 (18), 201 (100), 149 (10), 123 (15), 109 (14), 95 (23); HREIMS *m/z*: 522.3363 (calcd for C_33_H_46_O_5_, 522.3345).

20α-Hydroxy-6β-acetonyl-tingenol (**4**). Amorphous solid; [α]_D_^20^ + 8.4 (*c* 0.12, CHCl_3_); UV (EtOH) λ_max_ (log є) 220 (3.8), 284 (4.4) nm; IR ν_max_ 3620, 3218, 2950, 1708, 1462, 1374, 1290, 1204, 756 cm^−1^; ^1^H RMN (CDCl_3_, 500 MHz) δ 2.18 (3H, s, Me3′), 2.49 (1H, dd, *J* = 10.9, 17.0 Hz, H1′), 2.75 (1H, dd, *J* = 2.0, 17.0 Hz, H1′), 3.32 (1H, OH29), 5.47 (1H, OH), 5.79 (1H, OH), for other signals, see Table 1; ^13^C NMR (CDCl_3_, 125 MHz) δ 30.7 (q, Me3′), 52.0 (t, C1′), 208.2 (s, C2′), for other signals, see Table 1; EIMS *m/z* (%) 494 (M^+^, 7), 437 (4), 436 (5), 253 (12), 243 (18), 241 (32), 215 (28), 201 (100), 123 (16), 109 (44), 95 (18); HREIMS *m/z* 494.3342 (calcd for C_31_H_42_O_5_ 494.3316).

22-Deoxo-cognatine (**6**). Amorphous solid; [α]_D_^20^ + 51.2 (*c* 0.64, CHCl_3_); UV (EtOH) λ_max_ (log є) 308 (3.7), 267 (3.3), 217 (3.9), 211 (4.1) nm; IR ν_max_ 3391, 2925, 2855, 1724, 1646, 1574, 1455, 1294, 1051, 757 cm^−1^; ^1^H NMR (CDCl_3_, 500 MHz) δ 3.72 (3H, s, OMe3), 3.79 (3H, s, Ome29), 6.18 (1H, s, OH-2); for other signals, see Table 1; ^13^C NMR (CDCl_3_, 125 MHz) δ 52.2 (q, OMe29), 61.1 (q, OMe3), 68.1 (d, C21), for other signals, see Table 1; EIMS *m/z* (%) 508 (M^+^, 100), 493 (32), 490 (26), 475 (19), 461 (8), 447 (6), 415 (12), 377 (21), 361 (18), 349 (9), 325 (7), 309 (19), 295 (6), 283 (6), 257 (11), 231 (12), 217 (6) HREIMS *m/z* 508.2855 (calcd for C_31_H_40_O_6_ 508.2825).

### 3.4. Biological Assays

#### 3.4.1. Human Cancer Cell Lines

Cells used in this study were donated to the group by partner institutions. For screening, we used the following human solid tumour cell lines: A549 and SW1573 (non-small cell lung), HBL-100 and T-47D (breast), HeLa (cervix), and WiDr (colon). Cells were grown in RPMI 1640 medium supplemented with 5% FBS and 2 mM L-glutamine. Cells were incubated in 60 mm Petri dishes at 37 °C, 5% CO_2_, and 95% relative humidity. The cell culture medium used was RPMI 1640 supplemented with 5% heat-inactivated FCS, 2 mM L-glutamine, 100 U/mL penicillin, and 0.1 mg/mL streptomycin. Cell cultures were passaged biweekly using 0.05% trypsin and maintained at low passage.

#### 3.4.2. Antiproliferative Assay

DMSO was added to each sample to prepare 10 mM stock solutions. The tests were performed using our implementation of the NCI60 protocol [29]. Cells were grown in monolayers in 96-well plates. The maximum test concentration was 25 µM, and the sample exposure time was 48 h. The standard anticancer drugs cisplatin (CDDP) and 5-fluorouracil (5-FU) were used as positive controls.

#### 3.4.3. Label-Free Continuous Live Cell Imaging

HeLa cells were seeded onto a 35 mm high glass-bottom μ-dish (IBIDI, Germany) at a density of 80,000 cells/dish. After 24 h, the growth medium was replaced with RPMI 1640 phenol red-free medium, and cells were treated with 25 μM of selected compounds for 20 h using the CX-A label-free cell imaging system (Nanolive S.A., Tolochenaz, Switzerland), and the status of cell populations was recorded every 10 min. The initial field of observation was selected considering a homogeneous distribution of cells (236 μm × 236 μm). After the acquisition, images were processed using Eve segmentation and analysis software (Nanolive S.A., Tolochenaz, Switzerland): version2.1 to evaluate cell content and morphology parameters (Eve Analytics). The measurements were obtained for each population at each time point for every treatment.

#### 3.4.4. Yeast Strains, Growth Conditions and Dose-Response Curves

The haploid BY4741 was the reference wild-type strain. The Δ*yap1* mutant was obtained from the Euroscarf collection. The Δ*rad9* Δ*rad52* double mutant and the Δ*yrs1* Δ*yrr1* Δ*pdr1* Δ*pdr3* quadruple mutant (ΔΔΔΔpdr) have been described before [11,32]. Strains were grown in the YEPD medium (1%, *w*/*v*, yeast extract, 2%, *w*/*v*, peptone, and 2%, *w*/*v*, dextrose) at 25 °C. Growth was measured as optical density at 620 nm (OD_620_). The growth inhibition dose-response curves were determined through a broth microdilution assay in 96-well plates [34]. The concentration ranged from 1 to 128 μM (1:2 serial dilutions). DMSO 1% (*v*/*v*) served as a reference for “no drug” control. The inoculum was 0.001 OD_620_ (∼2.5·10^4^ cells/mL). The growth was measured after 24 h and 48 h of incubation at 25 °C. The GI_50_ (concentration that inhibited growth by 50%) was calculated by fitting a four-parametric curve to the normalised growth to DMSO 1% *v*/*v*. The aggregate growth (area under the curve) was calculated as reported before [31]. Growth inhibition under hypoxic conditions and in glycerol was determined through a halo assay due to the caveats to setting up these assays in liquid and their lower growth rates compared to normoxic YEPD [11]. Normoxic YEPD (dextrose as the fermentable carbon source) defined the “fermentation and respiration” condition. The “fermentation only” condition was achieved by creating anoxic conditions for the YEPD plate using the AnaerocultTM A mini kit (Merck, #1.01611.0001), whereas the “respiration only” condition was setup by replacing dextrose with glycerol (a non-fermentable carbon source) under a normoxic atmosphere. The cell density on the plate surface was adjusted to ∼100 cells/mm^2^, and the amount spotted for each compound was 10 nmol (1 μL from a 10 mM stock). Halo assays were also used to double-check mutant sensitivity.

#### 3.4.5. Bacterial Strains, Growth Conditions and MIC Determination

The bacterial strains were *Escherichia coli* ATCC35218, *Enterococcus faecalis* ATCC29212, and *Staphylococcus aureus* NRS402 [33]. Bacterial strains were grown in the cation-adjusted Mueller-Hinton medium. A broth microdilution assay was used to obtain a dose-response growth profile. Drug concentration and experimental setup were like those described above for yeast. The minimum inhibitory concentration (MIC) was determined after 24 h as the minimum concentration that rendered less than 10% of the growth observed in DMSO 1% *v*/*v*.

## 4. Conclusions

The current study reports on our efforts to find new anticancer drug candidates. Therefore, a series of fifteen phenolic *nor*-triterpenes, including three of them new in the literature and three reported for the first time as natural products, isolated from the root bark of Celastraceae species were evaluated. Evaluation of six human tumour cell lines revealed that three compounds from this series, named 6-oxo-pristimerol, zeylasteral, and demethyl-zeylasteral, exhibited significant activity on all the cell lines tested, the latter being more effective than the known chemotherapeutic drugs, cisplatin (CDDP) and 5-fluorouracil (5-FU), used as positive controls. Continuous live cell imaging of HeLa cells proposes that selected compounds induce apoptosis. To gain insight into the mode of action of these compounds, their effect on a drug-sensitive *Saccharomyces cerevisiae* model was investigated. The results suggest that zeylasteral and 7α-hydroxy-blepharodol are the most potent compounds, though their toxicity is blocked by the drug resistance mechanisms, whereas 22-deoxo-cognatine disrupts the respiratory chain via a mechanism unrelated to reactive oxygen species. Moreover, dimethyl-zeylasteral represents the optimal balance between target potency and accessibility. A comprehensive SAR analysis suggests that an aldehyde at C4 and a carboxyl group at C20 in a pristimerin scaffold contribute to the anticancer effectiveness.

The present work reinforces the potential of Celastraceae species as a source of lead compounds, enhancing our understanding of the therapeutic potential of phenolic *nor*-triterpenes and deserving the future rational design of anticancer agents based on this scaffold.

## Figures and Tables

**Figure 1 ijms-25-09470-f001:**
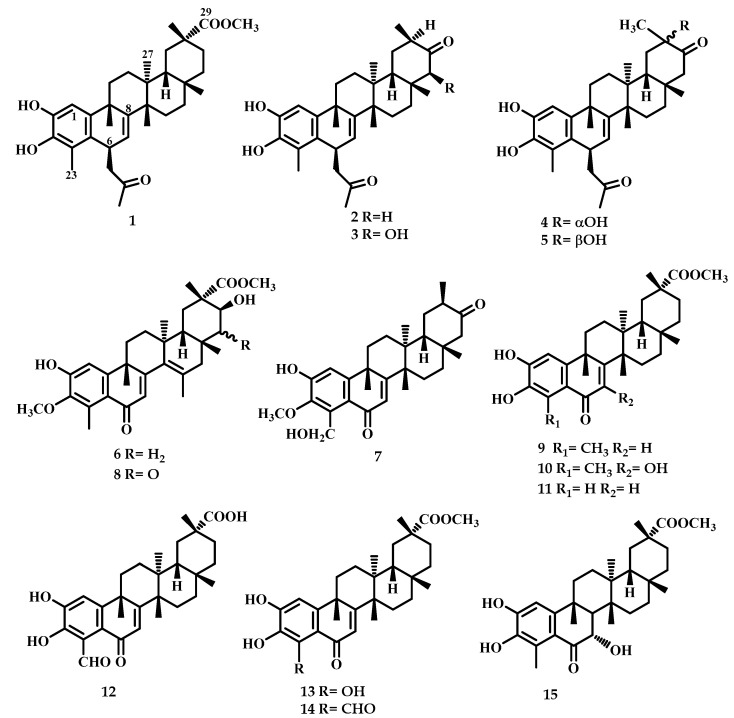
Structure of isolated phenolic *nor*-triterpenes (**1**–**15**).

**Figure 2 ijms-25-09470-f002:**
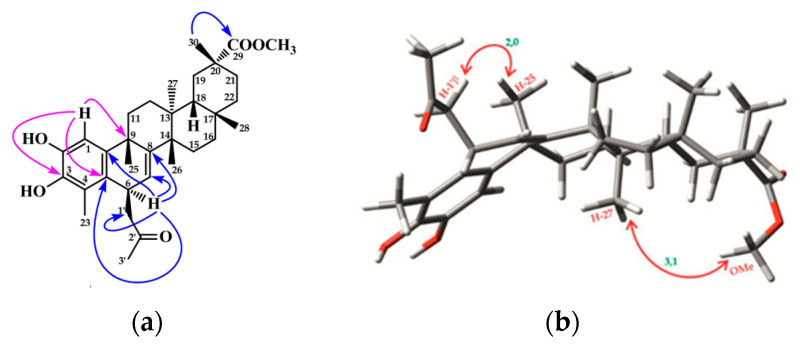
(**a**) ^1^H-^13^C long-range correlations HMBC and (**b**) ^1^H-^1^H ROESY (CDCl_3_, 500 MHz) experiments for **1**.

**Figure 3 ijms-25-09470-f003:**
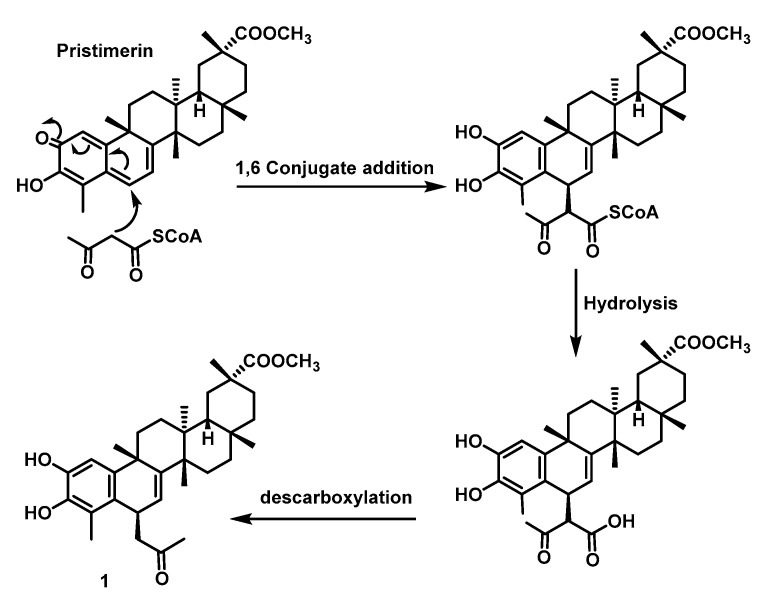
Plausible biosynthetic pathway for the phenolic *nor*-triterpene **1**.

**Figure 4 ijms-25-09470-f004:**
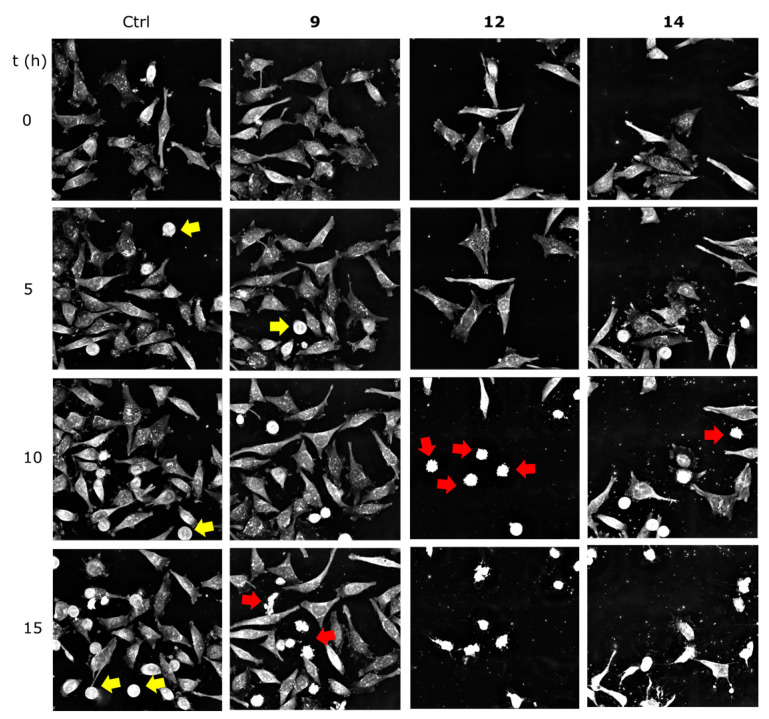
Representative snapshots of HeLa cells exposed to the selected compounds **9**, **12,** and **14** (25 μM, 20 h). Yellow arrows indicate cells undergoing mitosis. Red arrows indicate the presence of apoptotic cells.

**Figure 5 ijms-25-09470-f005:**
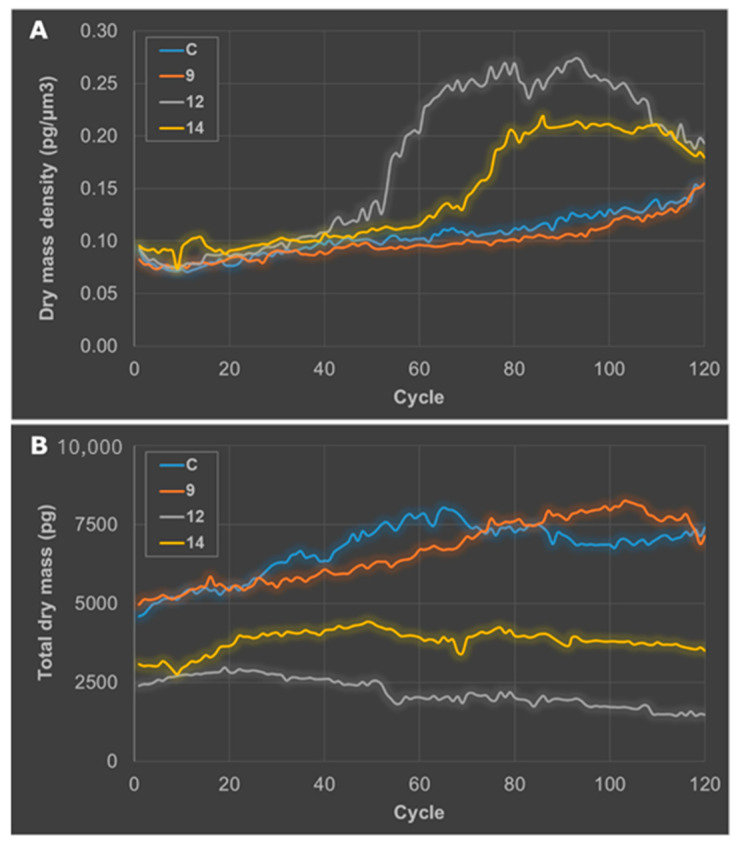
Kinetics of dry mass density (**A**) and total dry mass (**B**) of HeLa cells untreated or treated (25 μM, 20 h) with the three selected compounds (**9**, **12,** and **14**) over time.

**Figure 6 ijms-25-09470-f006:**
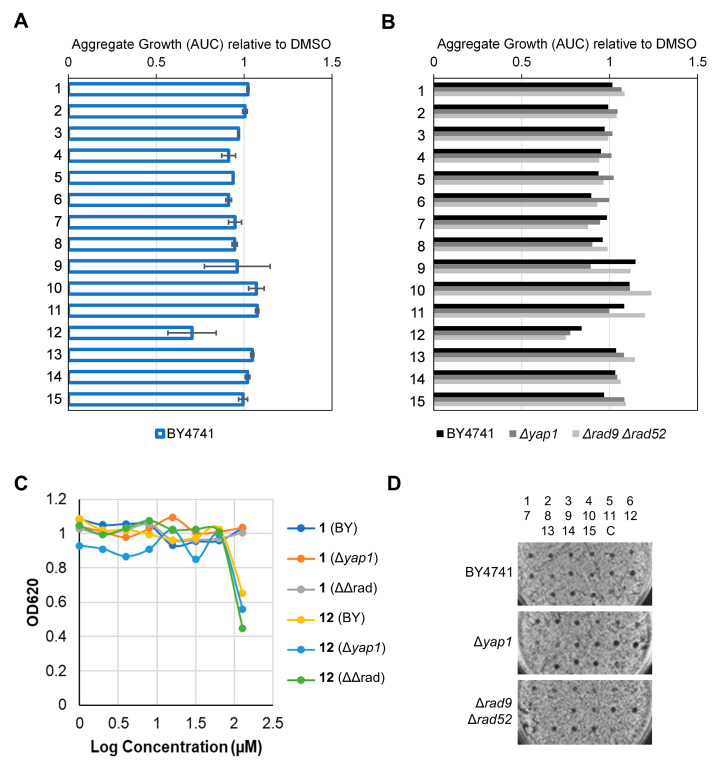
Compound sensitivity against yeast mutants for oxidative stress and DNA damage. (**A**) Relative aggregate growth (RAG) of the wild-type reference haploid strain BY4741. The aggregate growth covers the 1–128 µM range and has been normalised to DMSO (value = 1) (mean ± SEM, *n* = 2). (**B**) Comparison of RAG between the wild-type and its isogenic mutants defective for the oxidative stress response (Δ*yap1*) and the DNA damage repair (Δ*rad9* Δ*rad52*). A representative experiment is shown. (**C**) The 1–128 µM dose-response curve of (**B**) at 24 h. Note that partial inhibition is only seen at 128 µM, and there was no difference between the wild type and the mutants. (**D**) Halo inhibition assay for the same three strains after 48 h. The spots correspond to 1 nmol (1 µL from a 10 mM stock in DMSO). The position of the fifteen compounds is indicated above; C, 1 µL of DMSO.

**Figure 7 ijms-25-09470-f007:**
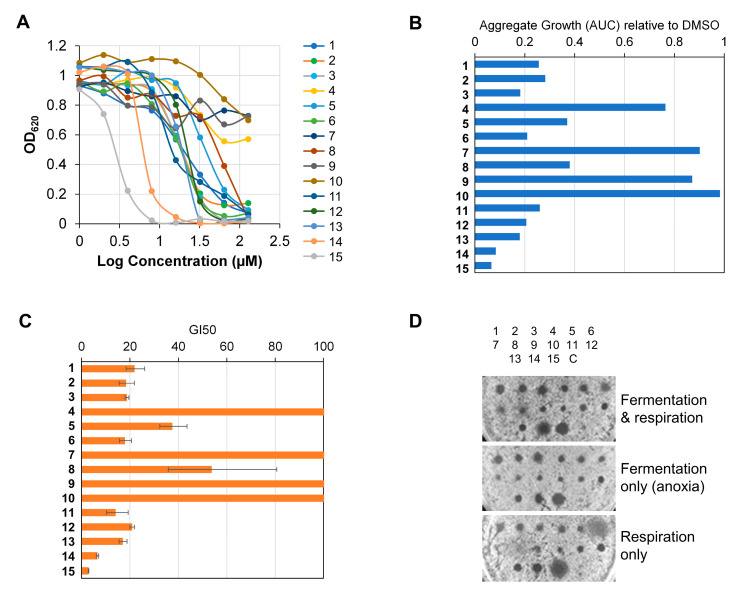
Compound sensitivity against a yeast strain defective in the pleiotropic drug resistance (ΔΔΔΔpdr). (**A**) A representative 1–128 µM dose-response curve after 24 h. (**B**) Relative aggregate growth (RAG) of (**A**). RAG is normalised to DMSO (value = 1). (**C**) GI_50_ of (**A**). Error bars correspond to a confidence interval of 95%. (**D**) Halo inhibition assay under different growth conditions (72 h). The spots correspond to 1 nmol (1 µL from a 10 mM stock in DMSO). The position of the 15 compounds is indicated above; C, 1 µL of DMSO.

**Table 1 ijms-25-09470-t001:** ^1^H (400 MHz) and ^13^C (100 MHz) NMR (δ, CDCl_3_, *J* in Hz in parentheses) data ^a^ of compounds **1**, **4,** and **6**.

No.	1		4		6	
	δ_H_	δ_C_ ^a^	δ_H_	δ_C_ ^a^	δ_H_	δ_C_ ^a^
1	6.81 s	109.2 d	6.80 s	109.2 d	6.92 s	110.2 d
2		142.3 s		142.2 s		151.6 s
3		140.3 s		140.3 s		144.7 s
4		119.8 s		120.0 s		134.2 s
5		128.1 s		127.8 s		122.0 s
6	3.99 ddd (2.5, 6.2, 10.4)	32.8 d	4.01 t (6.3)	32.8 d		186.1 s
7	5.85 d (6.2)	121.7 d	5.86 d (6.0)	121.6 d	6.00 s	128.8 d
8		150.1 s		149.6 s		161.5 s
9		37.2 s		37.0 s		41.2 s
10		142.0 s		142.3 s		151.9 s
11	1.80, 2.05 m	35.7 t	1.96 dt (5.3, 13.2) 2.07 m	35.7 t	1.94 dd (5.1, 13.4) 2.04 dd (3.4, 13.4)	37.4 t
12	1.53 m	30.4 t	1.52, 1.59 m	29.5 t	1.33 dd (3.4, 13.8) 2.46 td (5.1, 13.8)	35.7 t
13		37.7 s		37.9 s		35.2 s
14		43.8 s		42.9 s		134.9 s
15	1.40, 1.53 m	28.9 t	1.46, 1.71 m	28.5 t		127.5 s
16	1.82 m	36.7 t	1.43, 1.81 m	36.8 t	1.43 m 2.58 d (16.8)	38.9 t
17		29.7 s		34.6 s		42.4 s
18	1.49 d (8.0)	44.5 d	1. 69 m	43.8 d	1.48 m	43.3 d
19	1.63 dd (8.0, 16.0) 2.33 d (16.0)	30.8 t	2.21 m 2.28 dd (7.5, 16.8)	34.2 t	β 1.50 m α 1.89 d (10.9)	34.3 t
20		40.5 s		72.3 s		48.0 s
21	2.05 m	30.0 t		217.0 s	4.20 dd (5.4, 11.6)	68.1 d
22	0.99 m 2.01 m	34.8 t	2.16 d (19.4) 2.69 d (19.4)	47.7 t	1.55 m	43.1 t
23	2.18 s	11.2 q	2.18 ^b^ s	11.3 q	2.73 s	15.1 q
25	1.45 s	36.8 q	1.49 s	37.5 q	1.37 s	27.9 q
26	1.17 ^b^ s	22.3 q	1.27 s	22.4 q	1.72 s	21.7 q
27	0.52	18.2 q	0.59 s	16.5 q	0.87 s	24.0 q
28	1.05 s	31.6 q	1.28 s	24.9 q	1.23 s	31.3 q
29		179.3 s				178.9 s
30	1.17 ^b^ s	33.0 q	1.40 s	31.2 q	1.21 s	13.8 q

^a^ Data are based on DEPT, HSQC, and HMBC experiments. ^b^ Overlapping signals.

**Table 2 ijms-25-09470-t002:** Antiproliferative activity ^a^ (GI_50_ μM ± SD) ^b^ of phenolic *nor*-triterpenes **1**–**15**.

Compound	A549	HBL-100	HeLa	SW1573	T-47D	WiDr
**1**	24 ± 2.3	21 ± 7.7	24 ± 1.8	21 ± 7.7	>25	>25
**2**	>25	18 ± 6.6	10 ±6.0	17 ± 6.6	>25	>25
**3**	>25	12 ± 0.6	22 ± 4.5	19 ± 4.2	>25	14 ± 1.4
**4**	>25	18 ± 7.7	17 ± 5.7	21 ± 2.9	>25	>25
**5**	>25	>25	>25	>25	>25	>25
**6**	>25	>25	>25	>25	19 ± 7.9	>25
**7**	>25	>25	>25	>25	>25	>25
**8**	>25	>25	>25	>25	>25	>25
**9**	8.3 ± 3.5	3.1 ± 1.4	8.6 ± 0.2	4.0 ± 0.6	20 ± 6.7	8.1 ± 0.6
**10**	>25	>25	>25	23 ± 3.7	>25	>25
**11**	>25	>25	23 ± 3.6	>25	>25	>25
**12**	0.96 ± 0.36	0.45 ± 0.17	1.1 ± 0.2	1.2 ± 0.2	2.4 ± 1.0	5.8 ± 1.9
**13**	19 ± 6.5	>25	>25	11 ± 1.4	>25	>25
**14**	6.6 ± 1.8	3.4 ± 1.4	4.1 ± 0.9	3.0 ± 0.7	7.9 ± 1.1	9.9 ± 2.1
**15**	19 ± 6.8	20 ± 9.2	19 ± 8.2	12 ± 5.2	>25	>25
**CDDP ^c^**	4.9 ± 0.2	1.9 ± 0.2	1.8 ± 0.5	2.7 ± 0.4	17 ± 3.3	23 ± 4.3
**5-FU ^c^**	2.2 ± 0.3	4.4 ± 0.7	16 ± 4.5	3.3 ± 1.2	43 ± 16	49 ± 6.7

^a^ A549 (non-small cell lung), HBL-100 (breast), HeLa (cervix), SW1573 (non-small cell lung), T-47D (breast), and WiDr (colon) tumour cell lines. ^b^ Values represent mean ± standard deviation (SD) of at least three independent experiments. ^c^ CDDP (cisplatin) and 5-FU (5-Fluorouracil) were used as positive controls.

## Data Availability

Data are contained within the article and Appendix A.

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
