# Peer review of "Exploring the Anticancer Potential of Phenolic nor-Triterpenes from Celastraceae Species"

_ijms, 2024, doi:10.3390/ijms25179470_

Round 1
Reviewer 1 Report
Comments and Suggestions for Authors
The paper is very interesting. I think it is a valuable contribution and should be published.
I found only minor things I would suggest to the authors to change :
Line 22 - antitumoral entities – perhaps this can be replaced with: compounds with antitumor activity
Figure 6 - should be larger for easier reading.
Besides these minor issues, I have not found anything else that needs to be resolved.
Comments on the Quality of English LanguageI found only minor things I would suggest to the authors to change :
Line 22 - antitumoral entities – perhaps this can be replaced with: compounds with antitumor activity
Author Response
REVIEWER #1
The paper is very interesting. I think it is a valuable contribution and should be published.
I found only minor things I would suggest to the authors to change :
REMARK 1: Line 22 - antitumoral entities – perhaps this can be replaced with: compounds with antitumor activity
ANSWER: This point has been corrected in the revised version following the reviewer’s suggestion
REMARK 2. Figure 6 - should be larger for easier reading.
ANSWER: This point has been corrected.
Besides these minor issues, I have not found anything else that needs to be resolved.
Reviewer 2 Report
Comments and Suggestions for Authors
This research is of significant interest as it aims to discover new therapeutic agents against cancer. The potential for these new compounds to be more effective and less harmful to patients is promising.
The authors used plants used in traditional medicine to search for new anticancer compounds. To do so, they designed an appropriate experimental strategy that includes extraction, purification, and deep instrumental analysis to analyze 15 compounds found; based on the relevant analyses, they described three compounds, 9, 12, and 14, which they would study in different biological tests; they demonstrated the region of the compound that would have the desired biological activity. They described that apoptosis is the type of death induced by these compounds and that the compounds with the best antiproliferative activities are 12 and 14. However, the studies in the different strains of mutant yeasts did not find significant biological activity in the generation of DNA damage or oxidative stress.
The effects on oxidative stress or the respiratory chain were only found in strain ΔΔpdr, which indicates low or no activity in these metabolic pathways under normal conditions.
What is the mode of action in cell lines of human origin?
I would only have to ask, if they saw the effect in the tumor cell lines, why didn't they analyze oxidative stress and DNA damage in those lines?
Author Response
REVIEWER #2
This research is of significant interest as it aims to discover new therapeutic agents against cancer. The potential for these new compounds to be more effective and less harmful to patients is promising.
The authors used plants used in traditional medicine to search for new anticancer compounds. To do so, they designed an appropriate experimental strategy that includes extraction, purification, and deep instrumental analysis to analyze 15 compounds found; based on the relevant analyses, they described three compounds, 9, 12, and 14, which they would study in different biological tests; they demonstrated the region of the compound that would have the desired biological activity. They described that apoptosis is the type of death induced by these compounds and that the compounds with the best antiproliferative activities are 12 and 14. However, the studies in the different strains of mutant yeasts did not find significant biological activity in the generation of DNA damage or oxidative stress.
The effects on oxidative stress or the respiratory chain were only found in strain ΔΔpdr, which indicates low or no activity in these metabolic pathways under normal conditions.
REMARK 1: What is the mode of action in cell lines of human origin?
ANSWER: We showed that cell died through apoptosis (Figs 4 & 5). The molecular trigger is unknown and will require deepen and complex studies. It is likely that a eukaryotic-specific target is present from the lack of antibacterial activity (Table S1). Common modes of actions that trigger apoptosis without a protein-specific target are unlikely (but not impossible) based on the results obtained with yeast. Indeed, the whole set of compounds showed a profile suggesting neither DNA damage nor oxidative stress (Figs 6 and 7). This is now highlighted in the abstract (line 34).
REMARK 2: I would only have to ask, if they saw the effect in the tumor cell lines, why didn't they analyze oxidative stress and DNA damage in those lines?
ANSWER: As just stated, none of the active compounds against tumor cell lines gave a sensitivity profile in yeast compatible with DNA damage and/or oxidative stress. We used yeast to weigh up these common modes of action since it is simpler to evaluate in this model organism. In most instances, drugs that damage the DNA or induce ROS in yeast do so in humans as well. It is thus likely that our compounds exert their cell line toxicity through other(s) mechanism(s).
Incidentally, we have just noticed that the sole compound in which we found a signature compatible with respiratory chain disruption was misreferred in the MS. We have corrected this in the abstract and the conclusions (line 553). The compound was inactive against tumor cell lines though.
Reviewer 3 Report
Comments and Suggestions for Authors
The authors of the manuscript undertook to investigate the properties of new compounds isolated from Celastraceae species. In my opinion, the experiments were properly planned and performed. The results of the experiments were presented in a clear and legible manner. My suggestion is to include in the manuscript the results of studies examining the effect on apoptosis and antiproliferative activity of the isolated compounds against normal cell lines, which would allow for the exclusion/confirmation of their selective action.
Author Response
REVIEWER #3
The authors of the manuscript undertook to investigate the properties of new compounds isolated from Celastraceae species. In my opinion, the experiments were properly planned and performed. The results of the experiments were presented in a clear and legible manner.
REMARK 1: My suggestion is to include in the manuscript the results of studies examining the effect on apoptosis and antiproliferative activity of the isolated compounds against normal cell lines, which would allow for the exclusion/confirmation of their selective action.
ANSWER: Indeed, this is an important action we plan to include in future studies for the most potent compounds. We have included a statement of this limitation in the revised version (lines 262-3).